# UNDERSTANDING THE ROLE OF SPECTRAL SIGNAL IN UNSUPERVISED GRAPH DOMAIN ADAPTATION

## ABSTRACT

Unsupervised graph domain adaptation (GDA) addresses the challenge of transferring knowledge from labeled source graphs to unlabeled target graphs. However, existing methods primarily implement spatial message-passing operators, which are limited by the neglect of the unique roles of spectral signals in unsupervised GDA. In this paper, we initially conduct an experimental study and find that the low-frequency topology signals signify the shared cross-domain features, while the high-frequency information indicates domain-specific knowledge. However, how to effectively leverage the above findings persists as a perplexing conundrum. To tackle this issue, we propose an effective framework named Synergy Low-High Frequency Cross-Domain Network (SnLH) for unsupervised GDA. Specifically, we disentangle the low- and high-frequency components in the original graph, extracting global structures and local details to capture more discriminative information and enhance the graph-level semantics. For the low-frequency components, we design an optimization objective to maximize the mutual information among low-frequency features, promoting the model to learn more generalized low-frequency information. To further mitigate domain discrepancy, we introduce high-frequency information cross-domain contrastive learning to impose constraints on the domains. By effectively leveraging both low and high-frequency information, the learned features turn out to be both discriminative and domain-invariant, thereby attaining effective cross-domain knowledge transfer. Extensive experiments demonstrate the superiority and effectiveness of the proposed framework across various state-of-the-art unsupervised GDA baselines.

## 1 INTRODUCTION

Graph data has been widely applied in various fields due to its ability to naturally express complex relationships in the real world, such as social network analysis (Fan et al., 2019), drug discovery (Abbasi et al., 2019; Bongini et al., 2021), and traffic flow prediction (Li & Zhu, 2021). In real-world applications, graph data from different domains typically encounters the issue of domain shift (Wu et al., 2022b). As a result, graph domain adaptation (GDA) methods have emerged (Ding et al., 2018), aiming to transfer knowledge from the source domain to the target domain (Lin et al., 2023; Liu et al., 2023). GDA effectively alleviates the challenges posed by differences in data distribution during cross-domain learning of complex graph-structured data. However, traditional graph domain adaptation methods (Qiao et al., 2023) typically rely on supervised learning and fail when there is a severe lack of labeled data in the target domain. Data with rich labels is scarce or difficult to obtain in real situations and it always takes a lot of effort and costs to have a little. Therefore, unsupervised graph domain adaptation (UGDA) is proposed to address the above issue. The advantage of unsupervised graph domain adaptation is that it can learn an effective cross-domain transfer model without any labeled data from the target domain. UGDA addresses the limitation of traditional GDA which relies on labeled data and enhances the model's generalization ability across different graph domains. Among existing UGDA works, adversarial learning-based methods (Wu et al., 2020; Zeng et al., 2024) attempt to reduce distribution differences between domains through adversarial training. However, this strategy is limited by cross-domain feature differences and performs poorly on the target domain. Graph neural network-based methods (Yin et al., 2023) aim to align the distribution of domain data within the generated representation space. As it is difficult to learn a reliable representation, the alignment is not effective in some cases.

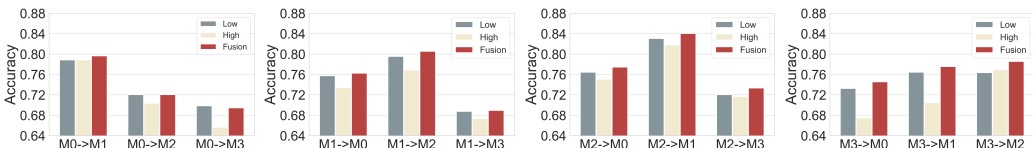

Figure 1: Impact of different spectral signals (*i.e.* low-frequency, high-frequency, and fusion-frequency) of domain information on Mutagenicity datasets. The experimental results show that both low-frequency and high-frequency signals have corresponding effects on the unsupervised cross-domain adaptation task.

Although these UGDA methods have made significant progress in their respective application scenarios, **one limitation** is that they typically use spatial domain operators to extract and align cross-domain features, relying heavily on *spatial information* while overlooking the impact of cross-domain features in the *spectral domain*. This results in the loss of specific information in the spectral domain, making it difficult to effectively align cross-domain features, leading to poor model performance in the target domain. An experimental analysis of the influence of spectral signal is shown in Figure 1. The graphs from different domains exhibit astonishing similarity in the low-frequency components of the same class of graphs in the spectral space while showing significant differences in the high-frequency components. This discovery prompted us to reassess the critical role of low- and high-frequency information in addressing cross-domain issues and to further explore the potential of low- and high-frequency information in unsupervised graph domain adaptation.

Inspired by the interesting findings above, we propose a novel and effective UGDA framework, named the synergy low-high frequency cross-domain network (SnLH). The framework concurrently optimizes both low-frequency and high-frequency information to alleviate the negative influence of domain dissimilarities in cross-domain circumstances. Specifically, we propose to use meticulously designed low- and high-frequency filters to separately extract low- and high-frequency information at the graph level for both the source and target domains, thereby enhancing the semantic representation of the graphs. For cross-domain low-frequency components, we employ a cross-domain mutual information constraint strategy to maximize the interaction between low-frequency information across domains, improving the model's ability to learn from cross-domain low-frequency information. However, relying solely on optimizing low-frequency information is insufficient to narrow and fully reduce domain differences. Therefore, for cross-domain high-frequency components, we introduce a cross-domain contrastive learning mechanism, aimed at finely tuning the differences in high-frequency components between positive and negative samples across domains, thereby strengthening the model's ability to distinguish cross-domain feature differences. Overall, our model captures and jointly optimizes both low- and high-frequency information across different domains, ensuring that the model learns representations that are both domain-invariant and discriminative. Our method has achieved significant improvements on benchmark datasets for graph domain adaptation through extensive experimental validation, outperforming existing methods. We summarize the contribution points as follows:

- We conduct an experimental study and discover some distinctive advantages of spectral signals in unsupervised graph domain adaptation tasks. Drawing inspiration from these findings, we have devised corresponding filters to extract low- and high-frequency information at the graph level from both the source and target domains. To the best of our knowledge, we are the first to study the spectral signal on the graph-level UGDA task.

- We propose a synergistic low-high frequency network (SnLH) that leverages cross-domain low and high-frequency information. By imposing constraints on low- and high-frequency information, SnLH effectively mitigates the impact of domain discrepancies on the model, enhancing its generalization capability on the target domain.

- Extensive experiments demonstrate that spectral domain information plays an important role in unsupervised graph domain adaptive tasks, our model achieves significant improvement on benchmark datasets and outperforms state-of-the-art baselines.

## 2 RELATED WORK

### 2.1 TYPICAL DOMAIN ADAPTION

Domain adaptation (DA), as a technique of transfer learning, is to improve the model's ability to generalize in scenarios with a different data distribution or label scarcity (Ben-David et al., 2006). However, real-world datasets often lack reliable labels (Achituve et al., 2021), making unsupervised domain adaptation a hot research topic. Current UDA methods are mainly divided into the following categories: The maximum mean discrepancy (MMD) method (Sun & Saenko, 2016) aligns the distributions by minimizing the MMD distance between the source and target features in a specific kernel space, but it fails to align the high-order distribution differences; adversarial-based methods (Ganin et al., 2016) distinguish the features of the source domain and the target domain through the domain discriminator. However the training process is unstable, which requires careful tuning of the discriminator and generator; the pseudo-label-based method (French et al., 2017) generates pseudo-labels for supervised learning of the target domain, while the quality of the pseudo-labels is difficult to guarantee, resulting in excessive deviation of the model training. These methods have undoubtedly achieved outstanding accomplishments in their respective fields, but there are still some unsolved challenges in these methods.

### 2.2 UNSUPERVISED GRAPH DOMAIN ADAPTATION

In practice, while existing unsupervised domain adaptation methods (Sun et al., 2017; Long et al., 2018) have achieved remarkable success in computer vision and natural language processing, there is a lack of unsupervised domain adaptation methods specifically designed for graph structure data because of the unique nature of graph structure data. Current unsupervised GDA approaches (Lin et al., 2023; Wu et al., 2022a) primarily focus on transferring information (Yin et al., 2022) from the source domain to the target domain using spatial operators of graph neural networks combined with domain alignment techniques (Luo et al., 2023). However, most of these methods (Guo et al., 2022; Zeng et al., 2024) overlook the significance of frequency domain information in graph domain adaptation (Yin et al., 2023). Therefore, in this paper we first explore the influence of frequency domain information and effectively leverage this knowledge to mitigate domain discrepancies, resulting in a significant improvement in the accuracy of the graph classification task.

## 3 PROBLEM DEFINITION AND PRELIMINARY

### 3.1 PROBLEM DEFINITION

Consider a graph $\mathcal{G} = \{\mathcal{V}, \mathcal{E}, X\}$ with the node set $\mathcal{V}$, the edge set $\mathcal{E}$ and $X \in \mathbb{R}^{N \times F}$ represents the feature matrix of the graph, where $F$ represents the feature dimension of each node. Let $A \in \mathbb{R}^{N \times N}$ be the adjacency matrix, $\widetilde{A} = D^{-\frac{1}{2}} A D^{-\frac{1}{2}}$ is the normalized adjacency matrix, where $N$ denotes the number of nodes, $D$ represents the degree matrix. For unsupervised graph domain adaptation, we initially define $\mathcal{D}_s = \{(\mathcal{G}_i^s, \mathcal{Y}_i^s)\}_{i=1}^{N_s}$ as the set of labeled source domain, where $\mathcal{Y}_i^s$ represents the graph labels of source domain and $N_s$ represents the number of graph in the source domain. Similarly, the target domain defined as $\mathcal{D}_t = \{\mathcal{G}_j^t\}_{j=1}^{N_t}$ contains $N_t$ unlabeled examples.

### 3.2 SPECTRAL DECOMPOSITION

Spectral Decomposition refers to the Eigenvalue Decomposition of a matrix as the product of its eigenvalues and eigenvectors. For the Laplacian matrix $L$ or adjacency matrix $A$ of a graph, their spectral decomposition can be used to understand the topological properties of the graph. The spectral decomposition is as follows for the normalized graph Laplacian matrix $L^{sym}$.

$$L^{sym} = U\Lambda U^T \tag{1}$$

Here, $U$ is the eigenvector matrix of the Laplacian matrix, $\Lambda = diag(\sigma_1, \sigma_2, \ldots, \sigma_N)$ is the diagonal matrix, and the diagonal elements are the eigenvalues of the Laplacian matrix.

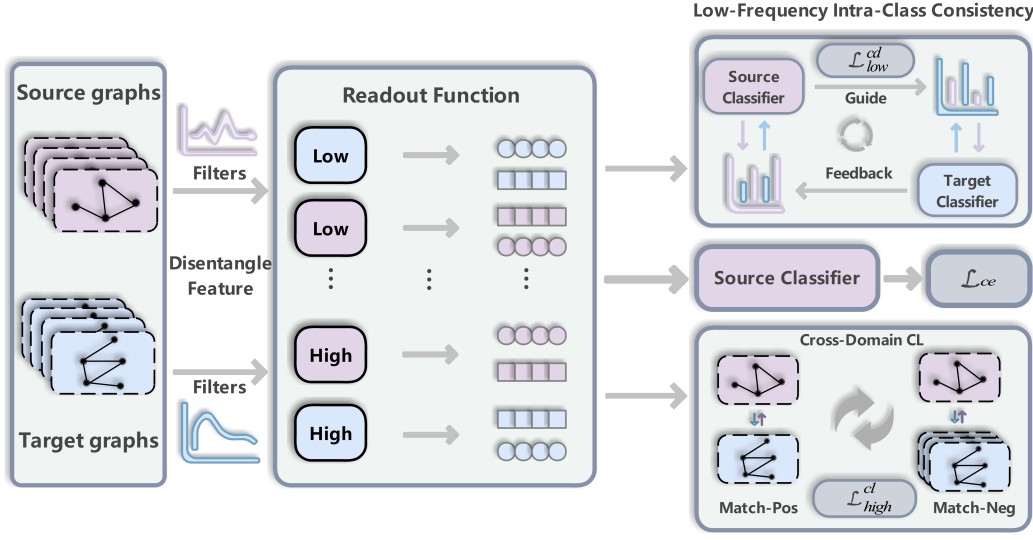

Figure 2: There are three modules in SnLH. Firstly, the low- and high-frequency information of the graph is extracted using the designed filter. Secondly, mutual information maximization of low-frequency information across different domains is processed and constrained. Lastly, high-frequency information from different domains undergoes cross-domain contrastive learning.

## 3.3 GRAPH SIGNAL PROCESSING

In graph signal processing (Shuman et al., 2013), the eigenvectors of the normalized Laplacian matrix can be regarded as a basis in the graph Fourier transform. For a given signal $x \in \mathbb{R}^N$, the graph Fourier transform and its inverse, denoted $\hat{x} = U^T x$ and $x = U\hat{x}$ respectively, are defined for a given signal $x$. Therefore, the convolution $\triangledown_{\mathcal{G}}$ between the signal $x$ and the convolution kernel $S$ is expressed as follows.

$$S_{\triangledown_{\mathcal{G}}} x = U\big((U^T S) \otimes (U^T x)\big) = U s_\theta U^T x, \tag{2}$$

where $U$ represents the orthogonal matrix of the symmetric normalized graph Laplacian matrix $L^{sym} = I_n - D^{-\frac{1}{2}} A D^{-\frac{1}{2}} = I_n - \widetilde{A}$ after spectral decomposition, $\otimes$ denotes the operation of element-wise multiplication of vectors and $s_\theta$ is the convolution kernel in the spectral domain.

## 4 METHODOLOGY

In this section, we explore the problem of UGDA from a spectral domain perspective and propose a novel model called synergy low-high frequency cross-domain network. Specifically, inspired by the characteristics of low- and high-frequency graph information in cross-domain tasks, we designed a low and high-frequency filter that disentangles the semantic information of the graph into low and high-frequency components. To fully exploit these information, we imposed a mutual information maximization constraint on cross-domain low-frequency information and enhanced the model's ability to distinguish cross-domain feature differences through a cross-domain high-frequency contrastive learning mechanism. In the following, we will introduce each module in detail.

### 4.1 LOW-HIGH FREQUENCY SIGNAL DISENTANGLEMENT

Inspired by the significance of both low-frequency and high-frequency information for cross-domain adaptation tasks which is found in previous experimental study, we design a low-pass filter $S_{low}$ and a high-pass filter $S_{high}$ respectively. These filters serve to disentangle low-frequency and high-frequency information from the node features in graphs. By doing so, we aim to leverage these

distinct spectral components more effectively, which is crucial for enhancing the model's adaptability and performance when dealing with data from different domains:

$$S_{low} = \mu I_n + \widetilde{A} = (\mu + 1)I_n - L^{sym}, \tag{3}$$

$$S_{high} = \mu I_n - \widetilde{A} = (\mu - 1)I_n + L^{sym}, \tag{4}$$

where $\mu$ is a scaling ratio hyper-parameter constrained within the range of $[0, 1]$ and $I_n$ is the identity matrix. Furthermore, we utilize $S_{low}$ and $S_{high}$ as low- and high-frequency information extractors. The signal $x$ of each graph is then disentangled into two parts by the filter:

$$S_{low \nabla_{\mathcal{G}}} x = U f_{low} U^T x, \quad S_{high \nabla_{\mathcal{G}}} x = U f_{high} U^T x, \tag{5}$$

From the above equation, we can derive that the convolution kernel for the low-pass filter is denoted as $S_{low}$, and the high-pass filter is denoted as $S_{high}$, the convolution kernels in the spectral domain for the low- and high-frequency filters are $f_{low} = (\mu + 1)I_n - \Lambda$ and $f_{high} = (\mu - 1)I_n + \Lambda$, respectively. To obtain more valuable low- and high-frequency information, we naturally set $\mu = 1$, which can effectively capture the low- and high-frequency information from the source domain and the target domain. Furthermore, we can find from Equations 3 and 4 that the specific meaning of low-frequency information is the sum of node features and neighborhood features in the spatial domain, while high-frequency information represents the difference between node features and neighborhood features in the spatial domain.

According to the previous theoretical analysis, we can convert the convolution kernel in the spectral domain to the spatial domain to extract the graph information. Specifically, for the node features of each graph $X = \{x_0, x_1, \ldots, x_N\}$, We disentangle their features using filters:

$$l_v^{(k)} = \text{ReLU}\left(\mathbf{W}_{low}^{(k-1)} \cdot (S_{low} \cdot l_v^{(k-1)})\right), v \in \mathcal{V}, \tag{6}$$

$$h_v^{(k)} = \text{ReLU}\left(\mathbf{W}_{high}^{(k-1)} \cdot (S_{high} \cdot h_v^{(k-1)})\right), v \in \mathcal{V}, \tag{7}$$

where $l_v^{(k)}$ and $h_v^{(k)}$ represent low-frequency and high-frequency information of the $k$ layer, respectively, when $k = 0$, then $l_v^{(0)} = h_v^{(0)} = x_0$. To further obtain the graph-level feature representation, we pass the low- and high-frequency information through the readout function:

$$l_i = \text{Readout}(\{l_v^{(K)}\}_{v \in \mathcal{V}_i}), \quad h_i = \text{Readout}(\{h_v^{(K)}\}_{v \in \mathcal{V}_i}), \tag{8}$$

Here, $l_i$ denotes the low-frequency representation of the $i$-th graph, $h_i$ denotes the high-frequency representation of the $i$-th graph, and $\mathcal{V}_i$ denotes the set of nodes of the $i$-th graph. To ensure the generalization performance of the model on the target domain, we use the disentangled graph semantics to impose constraints $\mathcal{L}_{ce}$ on the model and optimize the model to improve its performance.

## 4.2 LOW-FREQUENCY INTRA-CLASS CONSISTENCY

Within the framework of cross-domain learning, our previous experiment has revealed a remarkable phenomenon: instances that share the same semantics exhibit an inherent consistency in their low-frequency feature space, while high-frequency features tend to carry more domain-specific information. This discovery has guided us in developing a model that can both capture cross-domain commonalities and flexibly adapt to within-domain differences. To effectively apply this characteristic to cross-domain tasks, we aim to let our model be predominantly guided by cross-domain low-frequency information, ensuring that the learning process of these common features remains unaffected by domain-specific variations.

**Source Low-Frequency Consistency**. Specifically, we leverage the abundant low-frequency supervision signals in the source domain to guide the model in learning the global consistency of cross-domain low-frequency information. To achieve this, we constrain the model by maximizing mutual information, ensuring that it learns a certain degree of global domain invariance.

$$D_{KL}\big(P_s(l^s) \parallel P_t(l^s)\big) = \sum_i P_t(l_i^s) \log \frac{P_t(l_i^s)}{P_s(l_i^s)}, \tag{9}$$

$$\mathcal{L}_{low}^s = \tau_{kd}^2 \cdot D_{KL}\big(P_s(l^s) \parallel P_t(l^s)\big), \tag{10}$$

Here, $D_{KL}(\cdot)$ represents the calculation of the divergence of two probability distributions. $P_s$ and $P_t$ represent the probability distributions of data in the source domain and target domain, respectively. The parameter $\tau_{kd}$ represents the temperature coefficient to soften the probability distribution of the model.

**Target Low-Frequency Consistency**. However, since our primary task is to ensure accurate classification in the target domain, constraining only the source domain information is insufficient. To achieve this goal, we need to constrain the target domain similar to the source domain low-frequency information $\mathcal{L}_{low}^t$, ensuring that the model not only performs well in the source domain, but also can effectively capture the cross-domain commonality, and promote the transfer between cross-domain low-frequency information. By doing so, we can enhance the model's adaptability in the target domain, leading to improved performance and robustness in downstream tasks, ultimately boosting overall generalization and prediction accuracy. The way of constraint is similar to that of constraining low-frequency information in the source domain.

Overall, our model not only focuses on information within the source domain but also effectively captures the critical information that exists in the target domain. Through this approach, we successfully bridge the gap between the source and target domains, allowing the model to transcend the limitations of a single-domain perspective. This process not only enhances the model's ability to generalize low-frequency knowledge across domains but also improves its performance in target domain tasks, enabling the smooth transfer of cross-domain low-frequency knowledge.

$$\mathcal{L}_{low}^{kd} = \mathcal{L}_{low}^s + \mathcal{L}_{low}^t, \tag{11}$$

### 4.3 HIGH-FREQUENCY CONTRASTIVE LEARNING

To further mitigate the impact of domain shift, relying solely on constraints on low-frequency features is far from sufficient. When dealing with cross-domain graph data, significant domain differences often lead to biased graph representations based on low-frequency information, directly affecting the model's performance on classification tasks in the target domain. Therefore, we employ contrastive learning on cross-domain high-frequency information to finely adjust the relative distances between positive and negative samples of high-frequency information across different domains, thereby enhancing the model's ability to recognize and handle cross-domain details. To achieve this goal, we benefit from constraining the cross-domain low-frequency information, allowing us to identify positive samples in the target domain that share the same semantics as those in the source domain. On this basis, we perform cross-domain contrastive learning, minimizing the relative distances between positive samples of high-frequency information with the same semantics across domains.

$$\mathcal{L}_{high}^{cl} = \sum_{i=1}^{N_s} \log \frac{s(h_i^s, h_i^t)}{\sum_{j=1}^{N_t} s(h_i^s, h_j^t)} + \sum_{i=1}^{N_t} \log \frac{s(h_i^t, h_i^s)}{\sum_{j=1}^{N_s} s(h_i^t, h_j^s)}, \tag{12}$$

where $s(\cdot, \cdot) = \exp(\cos(\cdot, \cdot)/\tau_{cl})$ and $h_i^t$ and $h_i^s$ are the positive of each other. Overall, for handling cross-domain high-frequency information, we employed a contrastive learning strategy that significantly enhances the model's sensitivity to and learning of domain differences. Through a carefully designed contrastive loss function, we strengthened the model's ability to recognize and encode domain-invariant features, deepened its understanding of domain diversity, and thereby achieved domain-invariant graph representations. We combine the classification loss of source domain with $\mathcal{L}_{ce}$ knowledge distillation of low-frequency information and contrastive learning of high-frequency information, resulting in the following overall loss function:

$$\mathcal{L} = \mathcal{L}_{ce} + \mathcal{L}_{high}^{cl} + \mathcal{L}_{low}^{kd}. \tag{13}$$

Overall, based on the discovery of the unique roles that low- and high-frequency information play in cross-domain tasks, we have ingeniously leveraged these information types to jointly optimize our model. Compared to existing methods, our approach sheds light on the significance of spectral signals in UGDA.

Table 1: Cross-domain graph classification result on Mutagenicity (source → target).

| Method | 0→1 | 1→0 | 0→2 | 2→0 | 0→3 | 3→0 | 1→2 | 2→1 | 1→3 | 3→1 | 2→3 | 3→2 | AVG. |
|---|---|---|---|---|---|---|---|---|---|---|---|---|---|
| WL subtree | 74.9 | 74.8 | 67.3 | 69.9 | 57.8 | 57.9 | 73.7 | 80.2 | 60.0 | 57.9 | 70.2 | 73.1 | 68.1 |
| GCN | 71.1 | 70.4 | 62.7 | 69.0 | 57.7 | 59.6 | 68.8 | 74.2 | 53.6 | 63.3 | 65.8 | 74.5 | 65.9 |
| GIN | 72.3 | 68.5 | 64.1 | 72.1 | 56.6 | 61.1 | 67.4 | 74.4 | 55.9 | 67.3 | 62.8 | 73.0 | 66.3 |
| GMT | 73.6 | 75.8 | 65.6 | 73.0 | 56.7 | 54.4 | 72.8 | 77.8 | 62.0 | 50.6 | 64.0 | 63.3 | 65.8 |
| CIN | 66.8 | 69.4 | 66.8 | 60.5 | 53.5 | 54.2 | 57.8 | 69.8 | 55.3 | 74.0 | 58.9 | 59.5 | 62.2 |
| CDAN | 73.8 | 74.1 | 68.9 | 71.4 | 57.9 | 59.6 | 70.0 | 74.1 | 60.4 | 67.1 | 59.2 | 63.6 | 66.7 |
| ToAlign | 74.0 | 72.7 | 69.1 | 65.2 | 54.7 | 73.1 | 71.7 | 77.2 | 58.7 | 73.1 | 61.5 | 62.2 | 67.8 |
| MetaAlign | 66.7 | 51.4 | 57.0 | 51.4 | 46.4 | 51.4 | 57.0 | 66.7 | 46.4 | 66.7 | 46.4 | 57.0 | 55.4 |
| DUA | 70.2 | 56.5 | 64.0 | 63.7 | 53.6 | 68.5 | 57.7 | 76.0 | 65.1 | 59.8 | 57.9 | 67.7 | 63.4 |
| DEAL | 76.3 | 72.6 | 69.8 | 73.3 | 58.3 | 71.2 | 77.9 | 80.8 | 64.1 | 74.1 | 70.6 | 74.9 | 72.0 |
| CoCo | 77.7 | 76.6 | **73.3** | 74.5 | 66.6 | 74.3 | 77.3 | 80.8 | 67.4 | 74.1 | 68.9 | 77.5 | 74.1 |
| To-UGDA | 78.6 | 75.7 | 73.1 | 75.7 | 61.2 | 62.3 | 80.3 | 83.5 | **79.7** | 73.3 | 72.7 | 75.6 | 74.3 |
| A2GNN | 57.3 | 54.2 | 58.6 | 54.5 | 55.5 | 55.5 | 54.7 | 54.4 | 57.3 | 55.4 | 57.3 | 54.7 | 55.8 |
| GALA | 76.4 | 69.6 | 70.0 | 63.2 | 58.4 | 60.6 | 76.9 | 80.1 | 65.7 | 66.5 | 65.6 | 70.6 | 68.6 |
| SnLH | **81.3** | **78.2** | 73.1 | **77.6** | **67.8** | **74.3** | **80.8** | **84.2** | 69.9 | **78.0** | **73.9** | **78.9** | **76.5** |
| | (±0.2) | (±0.4) | (±0.9) | (±0.2) | (±0.6) | (±0.8) | (±0.4) | (±0.3) | (±0.3) | (±0.5) | (±1.0) | (±0.5) | (±0.5) |

Table 2: Cross-domain graph classification result on NCI1 (source → target).

| Method | 0→1 | 1→0 | 0→2 | 2→0 | 0→3 | 3→0 | 1→2 | 2→1 | 1→3 | 3→1 | 2→3 | 3→2 | AVG. |
|---|---|---|---|---|---|---|---|---|---|---|---|---|---|
| WL subtree | 72.6 | 80.3 | 62.7 | 75.5 | 52.0 | 63.6 | 69.1 | 69.8 | **70.7** | 59.4 | **80.0** | 70.6 | 68.9 |
| GCN | 49.5 | 71.1 | 46.8 | 33.7 | 32.7 | 27.4 | 56.2 | 55.3 | 58.2 | 51.0 | 60.7 | 53.2 | 49.6 |
| GIN | 67.3 | 67.9 | 61.5 | 65.4 | 58.9 | 61.0 | 62.5 | 66.2 | 69.7 | 56.8 | 72.4 | 64.0 | 64.5 |
| GMT | 50.3 | 42.5 | 51.1 | 42.5 | 56.1 | 42.5 | 53.2 | 51.0 | 68.2 | 51.0 | 68.2 | 53.2 | 52.5 |
| CIN | 51.1 | 72.6 | 54.0 | 72.6 | 68.2 | 71.5 | 55.0 | 53.5 | 68.2 | 52.0 | 68.3 | 53.6 | 61.7 |
| CDAN | 59.6 | 73.8 | 56.7 | 73.7 | 71.2 | 73.2 | 55.5 | 57.3 | 69.9 | 54.6 | 69.8 | 56.6 | 64.3 |
| ToAlign | 51.0 | 27.4 | 53.2 | 27.4 | 68.2 | 27.4 | 53.2 | 51.0 | 68.2 | 51.0 | 68.2 | 53.2 | 50.0 |
| MetaAlign | 65.0 | 77.6 | 62.0 | 77.1 | 68.2 | 74.5 | 64.2 | 65.4 | 68.0 | 56.1 | 68.2 | 66.2 | 67.7 |
| DEAL | 65.6 | 73.0 | 58.0 | 71.6 | 60.1 | 73.1 | 62.8 | 65.0 | 65.8 | 53.9 | 57.6 | 56.7 | 63.6 |
| CoCo | 70.4 | 80.4 | 62.4 | 75.8 | 65.7 | 73.7 | 67.0 | 70.4 | 69.7 | 62.7 | 74.4 | 63.7 | 69.7 |
| To-UGDA | 55.9 | 73.5 | 55.0 | 72.4 | 67.9 | 73.0 | 55.0 | 56.5 | 63.5 | 53.4 | 66.2 | 56.1 | 62.4 |
| A2GNN | 60.5 | - | - | - | 39.3 | - | - | 39.5 | 39.3 | 39.5 | 60.7 | - | - |
| MTDF | 67.5 | 76.7 | **70.9** | **77.2** | **71.8** | **75.9** | 65.0 | 62.5 | **73.1** | 61.0 | 74.3 | 57.8 | 69.5 |
| SnLH | **73.3** | **81.4** | 65.6 | 77.1 | 67.5 | 74.8 | **70.3** | **71.9** | 70.4 | **63.6** | **76.4** | 70.4 | **71.9** |
| | (± 0.8) | (± 0.3) | (± 0.9) | (± 0.9) | (± 1.2) | (± 1.2) | (± 0.7) | (± 0.9) | (± 0.9) | (± 1.3) | (± 0.4) | (± 1.3) | (± 0.9) |

# 5 EXPERIMENTS

## 5.1 EXPERIMENTAL SETTINGS

**Datasets.** Our experiments are conducted experiments on several benchmark datasets from TU-Dataset, including Mutagenicity(M), NCI1(N), NCI109(N109), PROTEINS(P), DD(D), COX2(C), COX2_MD(CM), BZR(B), BZR_MD(BM). Following the partitioning method of (Yin et al., 2023), we divided Mutagenicity and NCI1 into four domains based on edge density: M0, M1, M2, M3 and N0, N1, N2, N3. The specific descriptions are as follows:

- **Mutagenicity:** This dataset focuses on the mutagenic properties of chemical molecules, with each graph representing a compound and a total of 4337 graphs.

- **NCI1 and NCI109:** These two datasets focus on screening for antitumor activity in different cell lines. NCI1 targets non-small lung cancer cell lines, while NCI109 targets ovarian cancer cell lines.

- **PROTEINS and DD:** These two datasets are related to protein structures. In PROTEINS, each graph is labeled to indicate whether the protein is an enzyme, intending to identify the protein's function. In DD, each graph is labeled to indicate whether the proteins form a stable dimer, to study interactions between proteins.

Table 3: Cross-domain graph classification result on PROTEINS, DD, COX2, COX2_MD, BZR, BZR_MD, NCI1, NCI109 (source → target).

| Method | P→D | D→P | C→CM | CM→C | B→BM | BM→B | N→N109 | N109→N | AVG. |
|---|---|---|---|---|---|---|---|---|---|
| WL subtree | 72.9 | 41.1 | 48.8 | 78.2 | 51.3 | 78.8 | - | - | - |
| GCN | 58.7 | 59.6 | 51.1 | 78.2 | 51.3 | 71.2 | 64.7 | 63.8 | 62.3 |
| GIN | 61.3 | 56.8 | 51.2 | 78.2 | 48.7 | 78.8 | 66.0 | 64.9 | 63.2 |
| GMT | 62.7 | 59.6 | 51.2 | 72.2 | 52.8 | 71.3 | - | - | - |
| CIN | 62.1 | 59.7 | 57.4 | 61.5 | 54.2 | 72.6 | - | - | - |
| CDAN | 59.7 | 64.5 | 59.4 | 78.2 | 57.2 | 78.8 | 69.5 | 61.3 | 66.1 |
| ToAlign | 62.6 | 64.7 | 51.2 | 78.2 | 58.4 | 78.7 | 67.6 | 65.2 | 65.8 |
| MetaAlign | 60.3 | 64.7 | 51.0 | 77.5 | 53.6 | 78.5 | 69.4 | 64.1 | 64.9 |
| DUA | 61.3 | 56.9 | 51.3 | 69.5 | 56.4 | 70.2 | - | - | - |
| DEAL | **76.2** | 63.6 | **62.0** | 78.2 | 58.5 | 78.8 | 71.3 | 65.8 | 69.3 |
| CoCo | 74.6 | 67.0 | 61.1 | **79.0** | 62.7 | 78.8 | 73.3 | 65.8 | 70.3 |
| To-UGDA | 59.3 | 66.7 | 51.2 | 75.6 | 61.9 | 79.2 | 75.7 | 69.9 | 67.4 |
| SnLH | 66.2 (± 0.8) | **70.1** (± 0.9) | 61.1 (± 1.6) | **79.0** (± 0.4) | **66.5** (± 2.4) | **80.4** (± 1.1) | **75.8** (± 0.9) | **76.9** (± 0.5) | **72.0** (± 1.1) |

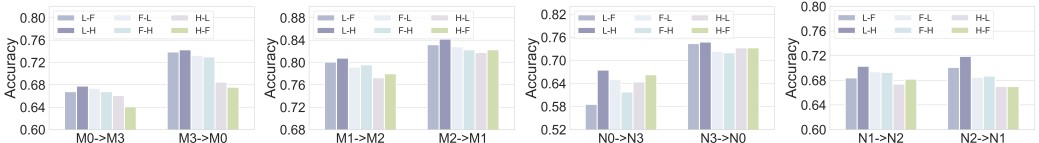

Figure 3: Influence of different low- and high-frequency information (*i.e.* L-F, L-H, F-L, F-H, H-L, and H-F) processing on Mutagenicity and NCI1.

- **COX2 and COX2_MD:** These two datasets are related to drug discovery, where each graph represents whether a compound has inhibitory activity against COX2 enzyme and COX2_MD enzyme activity.

- **BZR and BZR_MD:** These two datasets involve compounds' activity against the benzodiazepine receptor. In BZR and BZR_MD, each compound is labeled according to its activity against the benzodiazepine receptor.

**Baselines.** We compare the proposed model framework SnLH with multiple state-of-the-art methods, which include kernel-based approach: WL subtree (Shervashidze et al., 2011); graph neural network (GNN) based approaches: GCN (Kipf & Welling, 2016), GIN (Xu et al., 2018), CIN (Bodnar et al., 2021), GMT (Baek et al., 2021); domain adaptation approaches: CDAN (Long et al., 2018), ToAlign (Wei et al., 2021b), MetaAlign (Wei et al., 2021a); unsupervised graph domain adaptation approaches: DEAL (Yin et al., 2022), CoCo (Yin et al., 2023), To-UGDA (Zeng et al., 2024), A2GNN (Liu et al., 2024). The details of these methods are in Appendix B.

**Implementation details.** The number of layers of low- and high-frequency filters is set to 4. The learning rate is set to 2e-3, the embedding dimension of the hidden layer is set to 64, the temperature coefficient of distillation $\tau_{kd}$ is set to 2.0, and the temperature coefficient of cross-domain contrastive learning $\tau_{cl}$ is set to 0.2, and the ratio of mixed low- and high-frequency information $\lambda$ is set to 0.8. More details of the experiment can be found in Appendix C.

## 5.2 MAIN RESULTS

To validate the superiority and effectiveness of our proposed model, extensive experiments are conducted on the task of cross-domain graph classification. As seen from Table 1, 2, and 3, our method generally outperforms the current methods in the domain adaptation task under unsupervised conditions, and the average improvement across all these datasets is about 3 % compared to the best results in comparison algorithms. Specifically, our model gets the best performance in 10 tasks out of 12 on dataset Mutagenicity, 8 tasks out of 12 on dataset NCI1 and 6 tasks out of 8 on the other datasets. Accordingly, we can draw the following conclusions: (1) Most existing methods overlook the impact of spectral domain information, leading to a decline in model performance. Therefore,

Table 4: The results of ablation studies on Mutagenicity (source → target).

| Method | M0→M1 | M1→M0 | M0→M2 | M2→M0 | M0→M3 | M3→M0 |
|---|---|---|---|---|---|---|
| SnLH | **81.3 ± 0.2** | **78.2 ± 0.4** | 73.1 ± 0.9 | **77.6 ± 0.2** | 67.8 ± 0.6 | **74.3 ± 0.8** |
| w/o CL | 79.9 ± 0.5 | 76.7 ± 1.1 | **73.6 ± 1.3** | 76.5 ± 0.7 | 67.1 ± 0.9 | 73.6 ± 1.3 |
| w/o $KD_s$ | 79.6 ± 1.2 | 75.8 ± 1.4 | 72.6 ± 0.5 | 76.0 ± 0.8 | **68.4 ± 1.2** | 71.7 ± 0.8 |
| w/o $KD_t$ | 79.7 ± 0.7 | 75.9 ± 0.8 | 72.7 ± 0.4 | 76.8 ± 0.7 | 68.1 ± 1.1 | 73.4 ± 0.9 |
| w/o low | 79.1 ± 0.7 | 75.7 ± 0.6 | 72.4 ± 0.6 | 75.9 ± 0.3 | 67.6 ± 1.0 | 72.3 ± 0.7 |
| w/o high | 79.0 ± 0.7 | 73.8 ± 0.4 | 69.4 ± 0.6 | 74.3 ± 1.1 | 64.7 ± 1.2 | 66.9 ± 1.6 |
| repl. GCN | 77.4 ± 0.4 | 74.5 ± 0.3 | 71.9 ± 0.6 | 73.4 ± 0.5 | 65.9 ± 0.7 | 65.7 ± 2.8 |

Table 5: The results of ablation studies on Mutagenicity (source → target).

| Method | M1→M2 | M2→M1 | M1→M3 | M3→M1 | M2→M3 | M3→M2 |
|---|---|---|---|---|---|---|
| SnLH | **80.8 ± 0.4** | **84.2 ± 0.3** | **69.9 ± 0.3** | **78.0 ± 0.5** | **73.9 ± 1.0** | **78.9 ± 0.5** |
| w/o CL | 79.8 ± 0.7 | 83.1 ± 0.6 | 68.2 ± 0.7 | 76.6 ± 0.7 | 72.8 ± 0.4 | 78.8 ± 0.5 |
| w/o $KD_s$ | 80.5 ± 0.4 | 83.4 ± 0.4 | 67.8 ± 0.5 | 75.4 ± 0.8 | 71.8 ± 0.5 | 78.5 ± 1.0 |
| w/o $KD_t$ | 80.7 ± 0.3 | 83.3 ± 0.8 | 68.7 ± 0.7 | 76.8 ± 1.4 | 72.4 ± 1.0 | 77.6 ± 0.3 |
| w/o low | 79.7 ± 0.5 | 81.6 ± 0.6 | 67.4 ± 0.7 | 74.8 ± 0.5 | 72.5 ± 0.7 | 77.6 ± 0.5 |
| w/o high | 76.5 ± 0.4 | 81.2 ± 0.4 | 66.8 ± 0.4 | 70.8 ± 1.9 | 71.0 ± 0.3 | 75.7 ± 0.4 |
| repl. GCN | 77.1 ± 0.3 | 80.0 ± 0.3 | 64.9 ± 0.8 | 67.9 ± 2.4 | 69.5 ± 0.6 | 74.4 ± 0.5 |

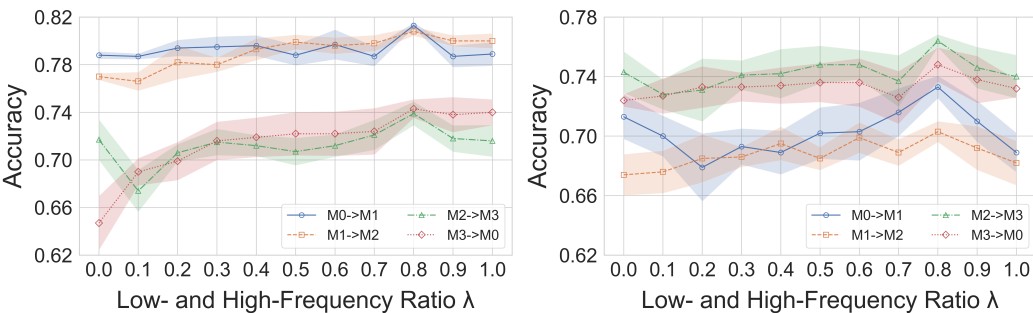

Figure 4: Sensitivity analysis on Mutagenicity and NCI1. We select eight predefined transfer tasks.

exploring the role of spectral domain information in cross-domain adaptation tasks is of significant importance. (2) Our method has better overall performance than the current GNN-based and adversarial methods, which not only confirms the positive influence of spectral domain information on cross-domain tasks but also shows that the use of low- and high-frequency information can further eliminate the domain shift on the model.

## 5.3 ABLATION STUDY

To evaluate the effectiveness of each crucial module, we introduce several variants of the model, which are represented as follows: (1) w/o CL: removing the cross-domain high-frequency information contrast module; (2) w/o $KD_s$: removing source domain mutual information module; (3) w/o $KD_t$: removing target domain mutual information module; (4) w/o low: removing low-frequency information extraction module; (5) w/o high: removing high-frequency information extraction module; (6) repl. GCN: using GCN for feature extraction instead of low- and high-frequency information. All parameter settings are the same as in the above experiments. The results of the ablation study are shown in Table 4, 5, and the following conclusions can be drawn: (1) The complete model SnLH outperforms all its variants, further verifying the importance of each module in unsupervised graph domain adaptation tasks. (2) Removing the mutual information maximization module of the source domain and the target domain separately, the performance decreases, indicating that the interaction between the cross-domain low-frequency information has a positive influence on the model. (3) When the cross-domain high-frequency contrastive learning module is removed, the per-

formance of the model decreases, indicating that cross-domain high-frequency information can help to improve classification accuracy. (4) Removing the extraction of low- or high-frequency information results in a significant performance drop, highlighting the importance of both low- and high-frequency information in cross-domain tasks. (5) Replacing low- and high-frequency information with features extracted by GCN leads to a notable performance decline, indicating that traditional low-frequency features cannot be effectively applied to cross-domain tasks.

### 5.4 INFLUENCE OF LOW- AND HIGH-FREQUENCY COMPONENT

In this subsection, to further illustrate the influence of low- and high-frequency information on the cross-domain task, we replace the input of the cross-domain mutual information maximization module and the cross-domain contrastive learning module with different signals, respectively. (1) L-F: Cross-domain low-frequency information is used for mutual information maximization constraint, and cross-domain mixed information is used for contrastive learning; (2) L-H: Cross-domain low-frequency information is used for mutual information maximization constraint, and cross-domain high-frequency information is used for contrastive learning; (3) F-L: Cross-domain mixed information is used for mutual information maximization constraint, and cross-domain low-frequency information is used for contrastive learning; (4) F-H: Cross-domain mixed information is used for mutual information maximization constraint, and cross-domain high-frequency information is used for contrastive learning; (5) H-L: Cross-domain high-frequency information is used for mutual information maximization constraint, and cross-domain low-frequency information is used for contrastive learning; (6) H-F: Cross-domain high-frequency information is used for mutual information maximization constraint, and cross-domain mixed information is used for contrastive learning. We tested and analyzed them on the datasets of Mutagenicity and NCI1. The experimental results are shown in Figure 3. The results show that low- and high-frequency information play different roles in the cross-domain task. Just as previous experimental study found for graph datasets from different domains, the low-frequency components of the same category of graphs in the spectral space show apparent similarities, while the high-frequency components show significant differences. This may be the core reason for the model to perform well in cross-domain tasks.

### 5.5 HYPERPARAMETER SENSITIVITY

In this subsection, to evaluate the influence of hyperparameter low- and high-frequency ratio $\lambda$ on model performance, we conduct experiments on Mutagenicity and NCI1 as shown in Figure 4. We limit the range of the hyperparameter $\lambda$ to [0-1] and fine-tune the hyperparameter with a span of 0.1. The experimental results show that when the parameter is gradually increased from 0.0 to 1.0, the overall accuracy of the model gradually increases and tends to be stable. When the proportion is small, high-frequency information accounts for the main part. According to our analysis, the reason is that high-frequency information cannot capture the global information between cross-domains well, which leads to the degradation of model performance. It shows that with the fusion ratio of low- and high-frequency information, and $\lambda$ is 0.8 (the low-frequency ratio is 0.8 and the high-frequency ratio is 0.2), the performance of the model reaches the best. At this time, the model can not only capture more valuable global information by virtue of cross-domain low-frequency information but also capture rich cross-domain local high-frequency information. Then, the contrastive learning module is used to eliminate the difference of cross-domain high frequency.

## 6 CONCLUSION

In this paper, we first conduct an experimental study and obtain interesting findings that low-frequency signals and high-frequency signals play different roles in cross-domain tasks and they both help to extract richer graph semantic information in cross-domain tasks. On this basis, we first design a low-frequency filter and a high-frequency filter to extract the low- and high-frequency information. To further use the low- and high-frequency information, we use the cross-domain mutual information constraint strategy to maximize the interaction between the cross-domain low-frequency information and perform contrastive learning on the cross-domain high-frequency information to fine-tune the high-frequency difference of the cross-domain information. Finally, we conduct extensive experiments on different benchmark datasets and compare them with various methods, our model outperforms the state-of-the-art methods. In future work, we will further explore the signifi-

cance of spectral signals on more complex graph-based tasks with the assistance of a large language model.

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

Appendix

# A Complexity Analysis

We represent the overall algorithmic flow of the model as follows. Furthermore, the time complexity of our model is analyzed.

## A.1 Time Complexity of Low-High Frequency Signal Disentanglement

For the low- and high-frequency filter module, the number of nodes $N$, the number of edges $|\mathcal{E}|$, the feature dimension $F$ of each graph, and the operation of each layer are considered when calculating the time complexity. For the operation of each layer, the time complexity is $O(|\mathcal{E}| + N \times F^2))$, and the model has $L$ layers, so the overall time complexity is $O\big(L \times (|\mathcal{E}| + N \times F^2)\big)$. It can be seen that the overall time complexity of the low- and high-frequency filter module is mainly related to the structure of the graph, that is, it is positively correlated with the number of nodes and the feature dimension.

## A.2 Time Complexity of Low-Frequency Intra-Class Consistency

For the part of low-frequency intra-class consistency, the calculation of time complexity mainly involves the number of samples $N_s$ and $N_t$ of the source domain and the target domain, and the number of classification classes $C$ of the task, and the overall time complexity is $O(N_s \times C + N_t \times C)$.

## A.3 Time complexity of high-frequency contrastive learning

For high-frequency contrastive learning, the computational time complexity mainly involves calculating the similarity matrix and the cyclic traversal to find positive and negative samples. For the number of source domain and target domain graphs are $N_s$ and $N_t$ respectively, the time complexity of computing the similarity matrix is $O(N_s \times N_t \times F)$, and the time complexity of cyclic traversal of positive and negative samples is $O\big((N_s + N_t) \times max(N_s, N_t)\big)$.

---

**Algorithm 1** The training process of SnLH model

---

**Input:** The labeled graph in the source domain $\mathcal{D}_s$; Unlabeled graph in the target domain $\mathcal{D}_t$.
**Output:** All the predicted values of the target domain graph along with the accuracy.
 1: Initialize the parameters of the model randomly.
 2: **while** the model is not convergence **do**
 3:    Sample batches of data from $\mathcal{D}_s$ and $\mathcal{D}_t$, respectively;
 4:    The sampled data is fed into a low- and high-frequency filter and a graph-level representation is obtained by a readout function;
 5:    Maximizing cross-domain low-frequency mutual information and contrastive learning of cross-domain high-frequency Information;
 6:    Calculate the overall loss function $\mathcal{L} = \mathcal{L}_{ce} + \mathcal{L}_{high}^{cl} + \mathcal{L}_{low}^{kd}$, and backpropagation, and update the model parameters.
 7: **end while**

---

# B Baselines

The baseline models for all comparisons are introduced as follows:

- **WL subtree:** The method is based on the Weisfeiler-Lehman algorithm, and the main idea is to construct the feature representation of a node by recursively aggregating the information of the node and its neighbors.

- **GCN:** The GCN model continuously updates the node information by aggregating the information of neighbors and uses an iterative way to generate coding vectors to capture cross-domain information.

- **GIN**: GIN is an architecture for graph neural networks that enhances graph representation by designing a specific aggregation mechanism that enables it to capture more complex graph structural information.

- **GMT**: GMT is a deep learning method for graph learning that combines the advantages of graph neural networks and Transformer architectures to enhance graph representation and matching accuracy.

- **CIN:** CIN aims to mitigate cross-domain differences by extending the traditional Weisfeiler-Lehman algorithm to handle fine-grained graph structures.

- **CDAN:** CDAN is a method for cross-domain learning, and its core idea is to reduce the distribution difference between the source domain and the target domain through conditional adversarial training.

- **ToAlign:** ToAlign is a deep learning method for cross-domain alignment, which aims to solve the feature distribution mismatch problem in the domain adaptation task.

- **MetaAlign:** MetaAlign is a meta-learning method for cross-domain adversarial learning, which aims to solve the feature alignment problem in domain adaptation.

- **DUA:** DUA is a cross-domain learning algorithm that improves the generalization ability of the model by considering the information of the source domain and the target domain at the same time, which aims to solve the problem of effective learning in the case of mismatched data distribution of the source domain and the target domain.

- **DEAL:** DEAL is an algorithm suitable for cross-domain learning, which uses adaptive perturbation and performs adversarial training with the domain discriminator to solve the problem of domain difference.

- **CoCo:** The CoCo method uses coupled branches and ensemble contrastive learning techniques to reduce the inter-domain differences and improve the performance of the model on cross-domain problems.

- **To-UGDA:** The TO-UGDA method aims to solve the problem of insufficient labeled data in the target graph domain by combining domain invariant features, adversarial alignment, and meta-pseudo-label techniques.

- **A2GNN:** The A2GNN model derives the generalization bound of multi-layer GNN and combines the constraint of maximizing the Mean difference (MMD) to reduce the difference between domains.

## C  EXPERIMENT DETAILS

In this part, we will further describe some experiment-related details as follows.

### C.1  MAIN RESULT DETAILS

In the main experiment, our hyperparameter settings are as follows: the ratio of low- and high-frequency information $\lambda$ is 0.8, the number of layers is 4, the dimension of the hidden layer is 64, the temperature coefficient of the cross-domain low-frequency mutual information maximization module $\tau_{kd}$ is 2.0, the temperature coefficient of the cross-domain high-frequency information contrast learning module $\tau_{cl}$ is 0.2, and the learning rate is 2e-3. Furthermore, we conducted several random experiments to obtain the mean and standard deviation of the output results as the final results. In the comparison experiment with the performance of the latest methods, the A2GNN model is mainly applied to the node classification task. To make a fair comparison, we processed the node feature output of A2GNN with the same processing as our model through the readout function, but the result is not ideal and cannot extract good graph representations.

### C.2  ADDITIONAL EXPERIMENTAL DETAILS

For the experimental study and the experiment of low- and high-frequency information influence, we conduct multiple experiments and record the average of the results as the final result. For the sensitivity analysis of the ratio parameter $\lambda$ of low- and high-frequency information, we make several experiments and record the mean and standard deviation as our final results.

