# OpenReview forum: "Understanding the Role of Spectral Signal in Unsupervised Graph Domain Adaptation"
_ICLR.cc/2025/Conference — ICLR 2025 Conference Withdrawn Submission_

### Official Review · Reviewer_ZBMk · 2024-10-27

**Soundness:** 3
**Presentation:** 2
**Contribution:** 2
**Rating:** 3
**Confidence:** 4

**Summary:**

This work separates graph data into low- and high-frequency components and applies specialized processing techniques: maximizing mutual information for low-frequency consistency across domains and using contrastive learning for high-frequency components.

**Strengths:**

Separating low- and high-frequency signals for UGDA introduces an innovative approach to better capture cross-domain information.

**Weaknesses:**

1. The idea of separating low- and high-frequency information is not novel, like [1][2]. Although these work faces different tasks, the core idea of guiding the model in learning the low-frequency and high-frequency information separately is the same.
2. It lacks new and related baselines, like [3].
3. Writing can be improved. For example, the first paragraph in the Introduction is too long. You should talk about graph data and graph domain adaptation in two different paragraphs. Besides, some words are too long, like
4. The use of mutual information and contrastive learning with frequency-based filters may add significant complexity, making the method harder to implement. Scalability on very large graphs with complex structures remains uncertain. You should provide computational complexity analysis or runtime comparisons on larger graph datasets.

[1] Bo D, Wang X, Shi C, et al. Beyond low-frequency information in graph convolutional networks[C]//Proceedings of the AAAI conference on artificial intelligence. 2021, 35(5): 3950-3957.
[2] Chen J, Lei R, Wei Z. PolyGCL: GRAPH CONTRASTIVE LEARNING via Learnable Spectral Polynomial Filters[C]//The Twelfth International Conference on Learning Representations. 2024.
[3] Luo J, Gu Y, Luo X, et al. GALA: Graph Diffusion-based Alignment with Jigsaw for Source-free Domain Adaptation[J]. IEEE Transactions on Pattern Analysis & Machine Intelligence, 2024 (01): 1-14.

**Questions:**

See weakness.

---

> ### Author Response · Authors · 2024-11-23
>
> Thanks for your thorough review and valuable suggestions. We address each point individually below.
>
> Q1. First of all, as for the separation of low and high-frequency information you mentioned, in the article you pointed out, that although the low and high-frequency information is separated, no corresponding processing is further done, it is only a simple linear combination, and the characteristics of low and high-frequency information are not well used to complete the corresponding task. Secondly, our method discovers the characteristics of low and high-frequency information in the frequency domain in the graph-level domain adaptation for the first time, and based on this, we process the signal more effectively. We test it on multiple data sets and get satisfactory results.
>
> Q2. Thank you for your valuable comments. For the latest baseline you mentioned, we conducted an algorithm comparison experiment on the Mutagenicity dataset, and the experimental results have been corrected in the modified PDF.
>
> Q3. Thank you for your comments. We have made appropriate adjustments to the introduction according to your comments.
>
> Q4. For the complexity of the method, the time complexity analysis of the model is presented in the appendix. Secondly, our method can be applied to ultra-large-scale networks, but as far as we know, we have not found a corresponding public data set. If you have a suitable data set to share, we are more than happy to apply the model to the corresponding data set and look forward to the performance of the model.

---

> > ### Comment · Reviewer_ZBMk · 2024-11-25
> >
> > Thanks for the authors' reply. I will keep my score.

---

### Official Review · Reviewer_UXpJ · 2024-10-31

**Soundness:** 2
**Presentation:** 2
**Contribution:** 2
**Rating:** 3
**Confidence:** 5

**Summary:**

The paper presents a framework for unsupervised graph domain adaptation (GDA) through the introduction of the Synergy Low-High Frequency Cross-Domain Network (SnLH). It identifies gaps in existing methodologies, including utilizing spatial message-passing operators while neglecting the potential of spectral signals. The authors conduct an experimental study revealing that low-frequency topology signals correlate with shared cross-domain features, while high-frequency signals denote domain-specific knowledge. SnLH disentangles these frequency components, optimizing low-frequency features to maximize mutual information and employing high-frequency contrastive learning to address domain discrepancies.

**Strengths:**

1. The model performs well in most databases.
2. SnLH provides a spectral signal view in solving graph-level domain adaption problems.
3. This work notably highlights the spectral signal information discrepancy in graph-level DA.

**Weaknesses:**

1. Novalty is limited. The paper claims they first explore the influence of frequency domain information and effectively leverage this knowledge to mitigate domain discrepancies. However, [1] also highlights its issue in the 2023 of GDA.
2. Lack of theoretical analysis. This work mentions mutual information many times when using this method. I doubt the effectiveness of this approach in practical terms. I doubt whether its impact on GDA is significant unless they can prove that the performance improvement is due to the introduction of the mutual information method rather than other domain alignment methods.
3. Lack of innovative methods. Low-high-frequency signal and low-frequency interclass consistency are basically existing losses, and improvement is incremental.
4. Graph-level DA impact is limited. Most existing GDA methods focus on node-level tasks. Recent graph-level work needs to clarify the importance of solving graph classification tasks due to the lack of work on that.





[1] Pang, Jinhui, et al. "Sa-gda: Spectral augmentation for graph domain adaptation." Proceedings of the 31st ACM international conference on multimedia. 2023.

**Questions:**

Same as Weaknesses.

---

> ### Author Response · Authors · 2024-11-23
>
> Thanks for your thorough review and valuable suggestions. We address each point individually below.
>
> Q1. As for the paper you mentioned in SA-GDA, it mentioned that using low and high-frequency information to process cross-domain information is for node level, but it does not show that low and high-frequency information has the same properties for graph-level tasks, but our preliminary experiments in the paper confirm this point. Secondly, the processing of low and high-frequency information in SA-GDA is very simple. Through some linear combination methods, the low and high-frequency information is simply used to alleviate the node-level domain differences, but our model treats the low and high-frequency information separately, and better utilizes this feature to alleviate the graph-level domain differences.
>
> Q2. Thank you for your valuable comments. As for whether mutual information has a significant impact on the model, we will analyze its impact on the overall performance of the model in detail in the ablation experiment Table (Table 4).
>
> Q3. Thank you very much for your comments, although this loss function is common, it is new to try and apply for graph domain adaptation tasks.
>
> Q4. Thanks for your valuable comments, we further illustrate recent work on graph-level tasks, confirming its impact.

---

> ### Comment · Reviewer_UXpJ · 2024-11-25
>
> Thanks for the authors' reply. I still find it difficult to agree on some core aspects. Specifically, the filtering operations extract high-frequency and low-frequency signals from both $A$ and $X$ in the graph, yet the model's $readout$ function lacks a significant novel design. Consequently, the graph-level task design appears to be more incremental than groundbreaking. So, I will keep my score.

---

### Official Review · Reviewer_u2Ds · 2024-11-01

**Soundness:** 3
**Presentation:** 3
**Contribution:** 3
**Rating:** 6
**Confidence:** 2

**Summary:**

This paper studies the problem of Unsupervised graph domain adaptation and proposes a new method named Synergy Low-High Frequency Cross-Domain Network (SnLH) for unsupervised GDA. It decouples the low- and high-frequency components in the original graph, extracting global structures and local details to capture richer semantic information and enhance the graph-level semantics. Extensive experiments demonstrate the superiority and effectiveness of the method across various state-of-the-art unsupervised GDA baselines.

**Strengths:**

- The studied problem is interesting and important.
- The paper is well-organized and clearly written.
- The idea of incorporating graph spectral signals into GDA is quite interesting and effective.

**Weaknesses:**

- Why A2GNN is introduced in the baseline? Is this method for node classification? It seems to be a wrong citation as well.
- The paper lacks some recent SOTA baselines such as "Multi-View Teacher with Curriculum Data Fusion for Robust Unsupervised Domain Adaptation".
- How about the influence of different GNN encoders?
- I suggest that the authors include some comparisons of computation time.

**Questions:**

See above.

---

> ### Author Response · Authors · 2024-11-23
>
> Thanks for your thorough review and valuable suggestions. We address each point individually below.
>
> Q1. A2GNN itself is a node-level classification method, and we process it with the same readout function as our method in subsequent experiments so that it can handle graph-level tasks.
>
> Q2. For the baseline model you mentioned, we did a comparative experiment on the NCI1 data set for the first time, and the experimental results show that our model is superior to this method. The experimental results are in Table 2 of the modified paper.
>
> Q3. As you said, we experimented with GCN Encoder in the ablation experiment, but we did not further verify the experiment of other encoders. We will further improve other encoders in the subsequent work.
>
> Q4. Thank you for your valuable comments. We will further improve the comparison experiments in the time dimension in the future. Based on the current part of the experiments, we are better than part of the baselines in time.

---

> > ### Comment · Reviewer_u2Ds · 2024-11-23
> > **Thanks for your comment.**
> >
> > Thanks for your comment. I suggest authors make modifications in the pdf now since they can revise the draft.

---

### Official Review · Reviewer_Ktfi · 2024-11-02

**Soundness:** 3
**Presentation:** 3
**Contribution:** 2
**Rating:** 3
**Confidence:** 4

**Summary:**

The paper addresses unsupervised graph domain adaptation (UGDA) by proposing the Synergy Low-High Frequency Cross-Domain Network (SnLH), which leverages low- and high-frequency spectral signals to handle cross-domain data transfer. Through disentangling and optimizing low- and high-frequency information, SnLH aims to enhance generalization across domains without target labels. Experimental results indicate that SnLH achieves competitive or superior performance compared to state-of-the-art UGDA methods on multiple datasets.

**Strengths:**

1. The approach provides a unique take on UGDA by distinguishing between low- and high-frequency spectral components, addressing previously overlooked aspects of spectral signal impact in GDA.
2. SnLH exhibits strong empirical performance, surpassing several baselines across diverse datasets, demonstrating its robustness and versatility.
3. The authors implement cross-domain mutual information maximization for low-frequency signals and contrastive learning for high-frequency signals, showcasing a well-structured approach to utilizing spectral information.
4. "Experimental studies reveal that low-frequency topology signals represent shared cross-domain features, while high-frequency information reflects domain-specific knowledge" is an interesting and intuitively reasonable finding.

**Weaknesses:**

1. Equations 9 and 10 represent a KL-divergence loss, not mutual information, and therefore are not equivalent to mutual information maximization, as claimed by the authors.

2. The authors claim that maximizing mutual information ensures the model learns global domain invariance on low-frequency features. However, this claim is unsubstantiated, and a more robust demonstration is needed to support this point.

3. Clarification is required on how  $P_s$ and $P_t$ are expressed or estimated within the model.

4. The motivation for applying contrastive learning to high-frequency features is insufficiently developed. A demonstration is necessary to justify why minimizing relative distances is appropriate for graph domain adaptation.

5. The proposed method appears inconsistent with the authors' motivations. Initially, the authors argue that low-frequency features capture domain-shared information, while high-frequency features are domain-specific. However, both the contrastive learning on high-frequency features and the KL minimization on low-frequency features aim to align feature distributions to achieve domain invariance. This approach does not align with the authors' original intent to treat high-frequency and low-frequency features differently due to different intrinsic properties.

**Questions:**

see weakness

---

> ### Author Response · Authors · 2024-11-23
>
> Thanks for your thorough review and valuable suggestions. We address each point individually below.
>
> Q1. In equations 9 and 10, $P_s$ and $P_t$ represent the classifiers of the source domain and target domain respectively, and the corresponding probability distribution is obtained by the input of low-frequency information, further constrained by KL divergence to maximize the cross-domain mutual information.
>
> Q3. $P_s$ and $P_t$ represent the source domain and target domain classifiers, respectively, after these two classifiers, the corresponding output can be obtained
>
> Q4. High-frequency information represents the difference between domains. Using the cross-domain contrastive learning mechanism to align high-frequency information allows the model to distinguish and distinguish similar samples effectively. It allows the model to maintain robustness and sensitivity to high-frequency information, which ultimately makes the model better adapt to the feature distribution of the target domain.
>
> Q5. The purpose of high-frequency information processing here is to better allow the model to distinguish similar samples in the target domain. As found in the pre-experiment, high-frequency information represents domain-specific information.

---

> > ### Comment · Reviewer_Ktfi · 2024-11-23
> >
> > Thanks for the authors' reply. I will keep my score.

---

### Note · Authors · 2024-11-26

**Comment:**

Dear Program Chair, Senior Area Chairs, Area Chairs, and Reviewers,

We would like to withdraw our submitted manuscript titled "Understanding the Role of Spectral Signal in Unsupervised Graph Domain Adaptation" with the manuscript submission number 7537.

We sincerely appreciate the time and effort invested by the reviewers and the Chairs in evaluating our manuscript. The constructive feedback provided has been valuable to us.

Thank you for your understanding.

Sincerely, Authors.

**Withdrawal Confirmation:**

I have read and agree with the venue's withdrawal policy on behalf of myself and my co-authors.